# Mutation of *GmIPK1* Gene Using CRISPR/Cas9 Reduced Phytic Acid Content in Soybean Seeds

**DOI:** 10.3390/ijms231810583

**Published:** 2022-09-13

**Authors:** Ji Hyeon Song, Gilok Shin, Hye Jeong Kim, Saet Buyl Lee, Ju Yeon Moon, Jae Cheol Jeong, Hong-Kyu Choi, In Ah Kim, Hyeon Jin Song, Cha Young Kim, Young-Soo Chung

**Affiliations:** 1Department of Molecular Genetics, College of National Resources and Life Science, Dong-A University, Busan 49315, Korea; 2Biological Resource Center, Korea Research Institute of Bioscience Biotechnology (KRIBB), Jeongeup 56212, Korea

**Keywords:** soybean, phytic acid, genome editing, CRISPR/Cas9, sgRNA, *GmIPK1*

## Abstract

Phytic acid (PA) acts as an antinutrient substance in cereal grains, disturbing the bioavailability of micronutrients, such as iron and zinc, in humans, causing malnutrition. *GmIPK1* encodes the inositol 1,3,4,5,6-penta*kis*phosphate 2-kinase enzyme, which converts *myo*-inopsitol-1,3,4,5,6-penta*kis*phosphate (IP_5_) to *myo*-inositol-1,2,3,4,5,6-hexakisphosphate (IP_6_) in soybean (*Glycine max* L.). In this study, for developing soybean with low PA levels, we attempted to edit the *GmIPK1 *gene** using the CRISPR/*Cas9* system to *introduce* mutations into the *GmIPK1* gene with guide RNAs in soybean (cv. Kwangankong). The *GmIPK1* gene was disrupted using the CRISPR/Cas9 system, with sgRNA-1 and sgRNA-4 targeting the second and third exon, respectively. Several soybean *Gmipk1* gene-edited lines were obtained in the T_0_ generation at editing frequencies of 0.1–84.3%. Sequencing analysis revealed various indel patterns with the deletion of 1–9 nucleotides and insertions of 1 nucleotide in several soybean lines (T_0_). Finally, we confirmed two sgRNA-4 *Gmipk1* gene-edited homozygote soybean T_1_ plants (line #21-2: 5 bp deletion; line #21-3: 1 bp insertion) by PPT leaf coating assay and PCR analysis. Analysis of soybean *Gmipk1* gene-edited lines indicated a reduction in PA content in soybean T_2_ seeds but did not show any defects in plant growth and seed development.

## 1. Introduction

Soybeans (*Glycine max* (L.) Merr.) are one of the world’s most important crops, containing oils, proteins, carbohydrates, dietary fibers, vitamins, and minerals. Transgenic soybean, a genetically modified crop, occupies a large portion of genetically modified organism (GMO) fields for its food, nutritional, industrial, and pharmaceutical applications [1,2,3]. To date, various useful genes with desired traits have been introduced into soybeans using *Agrobacterium*-mediated transformation, in which transgenic soybean plants have been successfully produced using mature or immature cotyledon explants [4] and improved by incorporating an alternative explant derived from mature soybean seeds [5,6,7,8,9]. Moreover, the addition of various thiol compounds, including dithiothreitol, L-cysteine, and sodium thiosulfate, significantly enhanced transformation efficiency by reducing the oxidative stress that causes tissue browning or cell death in CN explants. The modified thiol compound method also positively affects organogenesis and shoot growth in the shoot pad [6,8,10,11].

With these improved transformation processes, the consequences of foreign gene introduction have resulted in unexpected results due to the disruption of endogenous plant genes or exogenous gene silencing. In particular, RNAi application to the individual gene of interest has silenced entire gene families [12]. Thus, more accurate methods using different strategies, such as genome editing, have been developed. Genome editing rapidly produced new agronomically desirable traits, such as high oleic acid content, improved plant architecture, and rapid domestication of wild phenotypes with disease resistance and stress tolerance [13,14,15].

*Myo*-inositol-1,2,3,4,5,6-hexakisphosphate (IP_6_), commonly known as phytic acid (PA), is a major phosphorus form in soybean and other plant seeds [16]. In most cereal seeds, PA occurs as phytates in protein bodies, which can strongly chelate cations such as calcium, zinc, magnesium, copper, and iron [17]. Phytase enzymes are a class of phosphatases that catalyze the sequential hydrolysis of PA to less phosphorylated *myo*-inositol derivatives and inorganic phosphates [18]. During seed germination, endogenous phytase enzymes are activated and hydrolyze PA. This releases bound mineral cations, stored phosphorus (P), and inorganic phosphorus utilized for seedling growth [16]. Monogastric animals cannot utilize PA because of their lack of phytase enzymes. Thus, undigested phytic acid phosphorous (PA-P) in animal waste (manure) is one of the major causes of environmental phosphorus pollution [19,20]. Therefore, there is considerable interest in generating cereal crops with low PA content, such as soybeans, rice, maize, barley, common beans, and wheat.

Most phosphorus (P) in soybean seeds is found in the form of phytate (65–85%), leading to a deficiency in available P in humans consuming soybean seeds [16]. Therefore, soybean meal is usually enhanced with supplemental mineral P or phytase enzymes to increase P availability [21,22]. Twelve genes predicted to encode enzymes in the PA metabolic pathway were identified in cereals. Mutations in these 12 genes through mutagenesis and genetic engineering have been studied as an approach to creating *lpa* (low phytic acid) mutants in cereals [23,24]. For example, Yuan et al. (2007) previously reported the two soybean *lpa* mutant lines, *Gm-lpa-*TW-1 (a 2 bp deletion in the third exon of *GmMIPS1*) and *Gm-lpa-*ZC-2 (a G-A point mutation of *Gmipk1*) generated through gamma irradiation and EMS treatment, exhibited PA reductions of 66.6 and 46.3%, respectively [25]. Additionally, the *Gm-lpa-*TW-1 mutant showed a significantly reduced field emergence rate, whereas the *Gm-lpa-*ZC-2 mutation did not negatively affect seed viability and yield traits.

Inositol-penta*kis*phosphate 1-kinase (IPK1) converts 1,3,4,5,6-penta*kis*phosphate (IP_5_) to PA and catalyzes the last step of PA biosynthesis. The soybean genome contains three *IPK1* homologs on chromosomes 14, 6, and 4 (*Glyma14g07880*, *Glyma06g03310*, and *Glyma04g03240* in *Glycine max* v1.1 [3]. *GmIPK1* (*Glyma14g07880*) was highly expressed in immature soybean seeds compared with the other two genes [26]. The *Gm-lpa*-ZC-2 mutant (*Gmipk1*) dramatically decreased the PA content by up to approximately 50% compared with wild-type seeds. Therefore, we attempted to disrupt *GmIPK1* via genome editing of the CRISPR/Cas9 system to reduce PA content in soybean. The creation of targeted mutation by genome editing is a more applicable and accurate way than mutagen treatment. In this study, we generated transgenic soybean plants with genome-edited *Gm**IPK1* sgRNA-1 and sgRNA-4, which target different sites in the soybean *GmIPK1* gene. CRISPR/Cas9-mediated genome editing technology with two guide RNAs decreased the levels of PA in T_2_ genetically engineered plant seeds. Our results suggest that genome editing can be a precise tool to produce new crops by decreasing targeted metabolites in plants.

## 2. Results and Discussion

### 2.1. Generation of Gmipk1 Gene-Edited Soybean Plants by CRISPR/Cas9 System

A previous study reported that *GmIPK1* (*Glyma14g07880*) was highly expressed in seeds compared with other homologous genes *Glyma06g03310* and *Glyma04g03240* in *Glycine max* L. var. Pusa 9712 [26]. Thus, we first examined their expression levels by qRT-PCR in the following organs: leaves, stems, roots, flowers, seed pods, and seeds at two vegetative stages (VC and V2) and three reproductive stages (R2, R6, and R7) of the Korean soybean cultivar Kwangankong. High-level expression of *GmIPK1* and its two homologs was detected in the stems, roots, and seed pots of the R6 (full seed) stage and the seeds of the R7 (beginning maturity) stage. The highest expression levels were observed in the stems of the R6 stage and the seeds of the R7 stage. In particular, *GmIPK1* exhibited a significantly higher expression level relative to the two homologs (Figure 1). This suggests that *GmIPK1* may play an important role in seed development and maturation. Previous studies also reported that *IPK1* genes from rice and wheat had high expression levels at the seed development and maturation stages [27,28]. As the expression level of *GmIPK1* (*Glyma14g07880*) was significantly higher in the seed development and maturation compared with *Glyma06g03310* and *Glyma04g03240*, *GmIPK1* was selected as a target gene for gene editing to develop low phytic acid soybean plants. To isolate *GmIPK1* homologs in Kwangankong, PCR was performed with gene-specific primers for *GmIPK1* (*Glyma14g07880*) using cDNA and genomic DNA as templates. The amplified *GmIPK1* cDNA sequence contained an open reading frame of 1371 bp, encoding a protein of 456 amino acid residues. Nucleotide sequence analysis showed that the amplified *GmIPK1* cDNA and genomic sequences in Kwangankong had 100% similarity to the *GmIPK1* sequences from the soybean *Wm82.a22.v1* genome on JGI Phytozome 13 (http://phytozome-next.jgi.doe.gov/, accessed on 25 June 2020) [29] (data not shown).

To obtain soybean genome-edited plants with a low concentration of PA, four target sites (referred to as target 1, target 2, target 3, and target 4) were selected with PAM sequence 5′-NGG-3′ at the 5′ or 3′ ends of the second, third, and fifth exons of *GmIPK1* (Figure 2A). The mutagenesis of soybean protoplasts was performed using 40% PEG-mediated transformation with pJY_GmU6-10_SpCas9_PPT^R^-sgRNAs (sgRNA-1, sgRNA-2, sgRNA-3, and sgRNA-4). Since the *GmU6-10* promoter has high transcriptional activity and mutation efficiency in soybean hairy roots and *Arabidopsis thaliana* [30], we used the *GmU6-10* promoter instead of the *AtU6* promoter to control *GmIPK1* sgRNA expression in soybean using the CRISPR/Cas9 system (Figure 2B). In soybean protoplasts, mutation efficiency was calculated as the percentage of indels detected at the Cas9 cleavage site based on targeted deep sequencing. Because sgRNA-1 and sgRNA-4 induced mutations (0.8% and 0.5% indel frequency, respectively), we selected these two sgRNAs for the subsequent soybean transformation procedure (Figure 3C).

Two CRISPR/Cas9 vector constructs, pJY_GmU6-10_SpCas9_*Bar*:*GmIPK1* sgRNA-1 (referred to as *GmIPK1* sgRNA1) and pJY_GmU6-10_SpCas9_*Bar*:*GmIPK1* sgRNA-4 (referred to as *GmIPK1* sgRNA4), were used for *Agrobacterium*-mediated transformation in the Korean soybean cultivar Kwangankong based on the half-seed method with minor modifications (Figure 2B). *Agrobacterium*-mediated soybean transformation was established in our laboratory by combining the cotyledonary-node (CN) method [4] with alternative mature soybean seeds. This protocol has enabled the production of stable transgenic soybean plants. Moreover, a few modifications, such as the addition of thiol compounds (a mixture of L-cysteine, sodium thiosulfate, and dithiothreitol) to the cocultivation medium, had a positive effect on soybean transformation efficiency [5,6,8,9]. The soybean seeds were inoculated by applying a high concentration of binary vector-containing *Agrobacterium* solution to a wounded target area (Figure 3A, lane a). Because of the 10 mg L^−1^ PPT reaction in the shoot induction stage, most primary shoots turned yellow and became necrotic. Elongated shoots with relatively large leaves affected healthy root formation, and the plants were well acclimatized in small pots and grown in the greenhouse (Figure 3A, lanes b–g). PPT leaf coating was performed to confirm herbicide resistance in leaves from nontransgenic (NT) and transgenic plants. Five days after the treatment, the leaves of the NT plants turned yellow and reacted to PPT, whereas the transgenic leaves showed resistance to herbicides (Figure 3A lane h). Based on the PPT leaf coating assay results, the transformation efficiencies were approximately 2.9% and 6.1% with *GmIPK1* sgRNA-1 and *GmIPK1* sgRNA-4, respectively, and, in total, 15 and 25 transgenic plants were produced, respectively (Figure 3B and Figure 2C).

### 2.2. Integration and Expression of Transgenes in Transgenic Soybean Plants

To confirm the integration of transgenes in transgenic soybean plants, genomic DNA was isolated from 12 healthy and well-grown *GmIPK1* sgRNA-1 T_0_ plants and 16 *GmIPK1* sgRNA-4 T_0_ plants and analyzed via PCR (Figure 4). PCR analysis of *GmIPK1* sgRNA-1 transgenic soybean plants showed that the *GmIPK1* region, the selectable marker (*Bar* gene), and the *SpCas9* gene were amplified in all 12 transgenic lines as 718 bp, 548 bp, and 4170 bp fragments, respectively (Figure 4A). The *GmIPK1* region, *Bar* gene, and *SpCas9* gene were also verified in all 16 transgenic lines from *GmIPK1* sgRNA-4 (Figure 4B). Successful amplification of the *GmIPK1* region (718 bp) must be from the endogenous *GmIPK1* gene.

The copy number of transgene insertions in transgenic soybean plants (T_0_) was determined by Southern blot analysis of transgenic lines with sufficient leaf samples using the *Bar* probe. Four transgenic lines (#6, #10, #12, and #13) seem to have a single insertion, and five transgenic lines (#2, #3, #4, #5, and #7) have multiple copies of the transgene from the result of *GmIPK1* sgRNA-1. Among *GmIPK1* sgRNA-4 transformants, seven transgenic lines (#1, #2, #5, #13, #20, #21, and #22) seem to have low copy number of the transgene (data not shown).

The expression levels of the transgenes were analyzed in 12 *GmIPK1* sgRNA-1 and 13 *GmIPK1* sgRNA-4 transgenic soybean plants (T_0_) using RT-PCR because of the availability of tissue samples (Figure 5). The *Bar* gene was expressed in all 12 *GmIPK1* sgRNA-1 transgenic lines (Figure 5A). As expected, the expression of *GmIPK1* was detectable in all plants, including the nontransformed plants. Expression of the *GmIPK1* and *Bar* genes was also detected in all 13 *GmIPK1* sgRNA-4 transgenic lines (Figure 5B). Because of plant culture conditions, we obtained leaf samples from nine mutant lines (T_0_) induced by sgRNA-1 and six mutant lines induced by sgRNA-4. Next-generation sequencing (NGS) was conducted with T_0_ genomic DNA (sgRNA-1 #2, #3, #4, #5, #6, #7, #10, #12, and #13; sgRNA-4 #1, #2, #3, #5, #20, and #21). They all exhibited mutations with various insertion/deletion (indel) patterns at the *GmIPK1* gene target site (Appendix A). The sgRNA-1 and sgRNA-4 induced indel mutations in *GmIPK1* at efficiencies ranging from 0.1 to 84.3% and from 0.3 to 82.5%, respectively. However, the sgRNA-4 *Gmipk1* #21 line had the highest indel frequency (82.5%) and two major indel patterns: a five base deletion and a single base (A) insertion. Thus, we could expect to easily obtain homozygote lines in the next generation (T_1_).

### 2.3. Selection of CRISPR/Cas9-Induced sgRNA-4 Gmipk1 Gene-Edited Line and Measurement of PA Content

To check the heritability of *Gmipk1* mutations, T_1_ seeds were collected from all sgRNA-1 or sgRNA-4 *Gmipk1* mutants. Four T_0_ plants (lines #4 and #12 for sgRNA-1 and lines #20 and #21 for sgRNA-4) with high indel frequencies were selected to obtain T_1_ seeds and analyze gene editing patterns in the T_1_ generation. Ten seeds from each self-pollinated T_0_ plant were grown in a growth chamber, and a total of 24 T_1_ plants were obtained because of successful germination (sgRNA-1 #4 (5 lines), #12 (3 lines); sgRNA-4 #20 (5 lines), #21 (11 lines)). Indel frequencies for sgRNA-1 *Gmipk1* and sgRNA-4 *Gmipk1* gene-edited plants (T_1_) displayed various patterns, with different percentages of total mutation reads. Consistent with the deep sequencing results of T_0_ plants, the progenies of T_1_ plants showed five base deletions in *Gmipk1* #21-2 line and a single base insertion (A) in *Gmipk1* #21-3 with 100% indel frequency (Figure 6A). Other *Gmipk1* #21 T_1_ lines (#21-1, #21-4, #21-5, #21-6, #21-11) showed two types of mutations, half and half percentages, similar to the *Gmipk1* #21 T_0_ lines. Although T_1_ lines derived from *Gmipk1* #4, #13 (sgRNA-1), and #20 (sgRNA-4) had a high percentage of indel frequency, it was difficult to obtain homozygous mutant lines because the patterns varied (data not shown). Thus, we used *Gmipk1* #21-2 and #21-3 mutant lines for further experiments in this study. Both *Gmipk1* gene-edited lines had a premature stop codon almost immediately after the mutation sites. However, the T_1_ plant growth of *Gmipk1* #21-2 and #21-3 was comparable to that of the NT controls (data not shown). Significant differences in plant height and morphological phenotype were not observed between *Gmipk1* #21-2 and #21-3 mutants compared with NT controls. Similarly, Yuan et al. (2012) reported that the *Gmipk1* mutant (*Gm-lpa*-ZC-2) displayed low PA content in seeds and had no adverse effects on seed viability and agronomic traits in soybean. Therefore, we also measured total and free phosphorus (Pi) levels in the *Gmipk1* #21-2 and #21-3 mutants to determine whether they exhibited a reduction in PA content. Significant decreases of 20.7% and 25.7% in PA content were observed in T_2_ seeds obtained from *Gmipk1* #21-2 and #21-3 mutant lines, respectively (Figure 6B). To check potential off-target events by sgRNA-4 in homologous genes *Glyma06g03310* and *Glyma04g03240*, the possible sgRNA-4 target sites were analyzed in the two homologous genes. Five nucleotide sequences were different from those of the sgRNA-4 target site in *Glyma14g07880* (Appendix A). Genomic DNA was extracted from the leaves of *Gmipk1* #21-2 and #21-3′s T_2_ generation (four plants generated by #21-2 and two plants generated by #21-3), and then the potential off-target sites were amplified by PCR with gene-specific primers for sequence analysis. No mutations were found in the two homologous genes of six *Gmipk1* gene-edited plants, thus indicating that sgRNA-4 targeted only *Glyma14g07880* but not in its homologs and induced mutagenesis of *GmIPK1* through the CRISPR/Cas9 system (Appendix A).

In this study, we demonstrated the efficient reduction in PA content in soybean seeds through CRISPR/Cas9 gene editing of *GmIPK1*. Recently, efforts have been made to develop low PA plants in soybean by silencing the regulatory genes involved in the biosynthesis of PA, such as *GmMIPS1* and *GmIPK2* [31,32]. *GmMIPS1* and *GmIPK2* genes encode _D_-*myo*-inositol-3-phosphate synthase 1 and inositol polyphosphate 6-/3-/5-kinase 2, respectively. Kumar et al. (2019) reported that RNAi triggered seed-specific silencing of *GmMIPS1* as the target gene exhibited a 41% reduction in phytate content without damaging growth and seed development. Similarly, Punjabi et al. (2018) reported that the disruption of *GmIPK2,* the upstream gene of *GmIPK1* in the PA biosynthetic pathway, as the target gene by RNAi in the seeds showed low PA levels, moderate accumulation of inorganic phosphate, and elevated mineral content in some transgenic lines. These results suggest that PA reduction in RNAi transgenic soybean seeds did not show any significant abnormal agronomic traits, including seed germination and development. In the present study, CRISPR/Cas9-mediated gene-edited soybean plants with the *GmIPK1* gene resulted in an approximately 25% reduction in PA content without affecting seed development and plant growth. Based on previous reports and our results, we speculate that mutations in two or three genes such as *GmMIPS1*, *GmIPK2*, and *GmIPK1* could reduce PA levels by around 40–50% in seeds without any growth and development defects. To further investigate the agronomical traits of *Gmipk1* gene-edited lines, we are currently growing them in the GMO field. After looking into yield components and germination rates of *Gmipk1* gene-edited plants, the effect of reduced PA will be clarified.

In this study, *GmIPK1* (*Glyma14g07880*) was highly expressed in the seed developmental stage compared with the other homologs, *Glyma06g03310* and *Glyma04g03240*, in cv. Kwangankong (Figure 1). Therefore, we first focused on *GmIPK1* (*Glyma14g07880*) to generate soybean plants with low PA levels. Because *Glyma06g03310* and *Glyma04g03240* had extremely high sequence identity (96.65%) at the nucleotide sequence level, and their sequence similarity with *GmIPK1* was 80.90% and 80.75%, respectively, it was difficult to design one specific sgRNA that targeted all the *GmIPK1* homologs. Therefore, as the next step to obtain gene-edited soybean plants with a much lower PA content, we are currently generating CRISPR/Cas9-mediated soybean gene-edited plants using one specific sgRNA that simultaneously targets the two *GmIPK1* homologs (*Glyma06g03310* and *Glyma04g03240*). Subsequently, it will be crossed with the *Gmipk1* #21-2 and #21-3 lines.

## 3. Materials and Methods

### 3.1. RNA Isolation and Quantitative Real-Time Reverse Transcription PCR (qRT-PCR)

Expression patterns of *Glyma14g07880*, *Glyma06g03310,* and *Glyma04g03240* were analyzed in soybean various tissues and organs at different development stages by qRT-PCR. Soybean samples were kindly provided by Dr. Man-Soo Choi (National Institute of Crop Science, Rural Development Administration, Wanju, Korea). Total RNAs were isolated from the tissues and organs of different growth stages of the Korean soybean cultivar Kwangankong using the Ribospin Plant (GeneAll, Seoul, Korea), and cDNAs were synthesized using the iScript cDNA Synthesis Kit (Bio-Rad, Hercules, CA, USA) according to the manufacturer’s protocols. Quantitative PCR was performed using BioFACT 2X Real-Time PCR Series (with SFCgreen I) (BioFACT, Daejeon, Korea) according to instructions for the CFX96 real-time PCR detection system (Bio-Rad, Hercules, CA, USA). The relative expression levels were calculated using the comparative cycle threshold method. The soybean PEP carboxylase gene (*GmPEPCo*) was used as the reference gene to normalize all data (q*GmPEPCo* forward primer 5′-CATGCACCAAAGGGTGTTTT-3′ and reverse primer 5′-TTTTGCGGCAGCTATCTCTC-3′). The primers of *Glyma14g07880*, *Glyma06g03310,* and *Glyma04g03240* were utilized to amplify the 200 bp fragment (q*Glyma14g07880* forward primer 5′-GACGCAGCTGACTGGGTTTA-3′ and reverse primer 5′-GTATGCGGACCACTTTCCCA-3′; q*Glyma06g03310* forward primer 5′-GTGTATTTCATTGACTTGGATTTGAAGCG-3′ and reverse primer 5′-TCTGTAACTGTAACAAATCCTGCTAAAACTC-3′; q*Glyma04g03240* forward primer 5′-GTGTATTTCATTGACTTGGATTTGAAGCG-3′ and reverse primer 5′-GTCATCTGTAAGAGCCTGCTAAATCTC-3′) [26,33].

### 3.2. Construction of Two Gmipk1 Genome Editing Vectors for Soybean Transformation

The *GmIPK1* (*Glyma14g07880*) cDNA was amplified using the first-strand cDNA of total RNA from the developing seeds of Kwangankong as a template for polymerase chain reaction (PCR) with *GmIPK1* primers (forward primer 5′-ATGGCATTGACTTTGAAAGAGG-3′ and reverse primer 5′-TCAATATGCAGCATTAGATGCCTT-3′). The PCR products were purified by gel purification using the Expin Gel SV kit (GeneAll, Seoul, Korea) and cloned into the All One PCR cloning vector (BioFACT, Daejeon, Korea). Finally, *GmIPK1* cDNA was confirmed by DNA sequencing and comparison with the *GmIPK1* sequences from the soybean *Wm82.a22.v1* genome on JGI Phytozome 13 (http://phytozome-next.jgi.doe.gov/, accessed on 25 June 2020) [29].

Target sequences were identified in the coding region of the *GmIPK1* gene to design unique sgRNAs. Four guide RNA sequences (sgRNA-1, sgRNA-2, sgRNA-3, and sgRNA-4) containing a protospacer adjacent motif (PAM) sequence were selected using the Cas-Designer site (http://www.rgenome.net/cas-designer/, accessed on 27 April 2021). To verify the efficiency of *Gm**IPK1* sgRNAs, we constructed a pJY_GmU6-10_SpCas9_*Bar* plasmid vector expressing *Streptococcus pyogenes Cas9* (*SpCas9*) and *Gm**IPK1* sgRNAs in a CRISPR/Cas9 system [34]. The *SpCas9* gene was expressed under the control of the *AtRPS5A* promoter in the pJY_GmU6-10_SpCas9_ *Bar* plasmid, and the soybean *U6-10* promoter (*GmU6-10*) was used to drive *Gm**IPK1* sgRNA expression in soybean plants [30]. Additionally, the *Bar* gene was used as a selectable marker gene for modified phosphinothricin (PPT).

Protoplasts were extracted from the unifoliate leaves of soybean plants by incubation with 3x valosin-containing protein (VCP) enzymes for 12 h at room temperature. PEG-mediated RNP delivery was performed as previously described [35] with minor modifications. Briefly, 2 × 10^5^ protoplasts were mixed with preassembled Cas9/gRNA (1:5 molar ratio) in 150 µL of MMg (4 mM MES, 0.4 M mannitol, and 15 mM MgCl_2_) via an equal volume of freshly prepared PEG solution (40% (*w*/*v*) PEG 4000, 0.2 M mannitol and 0.1 M CaCl_2_). Transfected protoplasts were kept at 22 °C for 2 days and collected for indel frequency analysis. The two desired plasmids, pJY_GmU6-10_SpCas9_PPT^R^:*Gm**IPK1* sgRNA-1 and pJY_GmU6-10_SpCas9_PPT^R^:*Gm**IPK1* sgRNA-4, were transformed into *Agrobacterium tumefaciens* EHA105 [36] and used for soybean transformation.

### 3.3. Next-Generation Sequencing (NGS) Analysis

Genomic DNA was isolated from the leaves of gene-edited T_0_ and T_1_ plants using the cetyltrimethylammonium bromide (CTAB) procedure. Targeted regions of *SpCas9* and sgRNA complexes were amplified by three rounds of PCR. First, the genomic region containing the guide RNA-binding sites was amplified to approximately 500 bp. First-time PCR products were diluted 1:10 and used as templates for the second PCR. Illumina adaptors and barcode sequences were added to the second and third PCR products. The final PCR products were quantified using a NanoDrop (Thermo Scientific, Waltham, MA, USA), and targeted deep sequencing was supported by Bio Core facilities in KAIST (http://biocore.kaist.ac.kr, accessed on 15 July 2021). Indel frequencies and mutation patterns were analyzed using a CAS-Analyzer in RGEN tools [37].

### 3.4. Agrobacterium-Mediated Soybean Transformation

Stable transgenic soybean plants were generated using mature soybean seeds of the Korean cultivar Kwangankong, following the half-seed method with modifications described previously [6,7]. Three batches of transformation experiments were carried out with 130–150 soybean seeds per batch for soybean transformation. The BASTA herbicide was used to coat the upper surface of two trifoliate leaves of transgenic (T_0_) plants with a mixture of 100 mg L^−1^ DL-phosphinothricin (PPT) and Tween-20 to screen putative transformants expressing the *Bar* gene. The response to the herbicide was observed 3–5 days after PPT application. Plants with undamaged and herbicide resistance were selected as putative transformants, continuously grown in a greenhouse, and the seeds were harvested.

### 3.5. Confirmation of Transgenes in Transgenic Soybean Plants (T_0_)

Genomic DNA was extracted from leaf tissues of nontransgenic (NT) and transgenic (T_0_) plants using CTAB. To detect transgenes in soybean plants, PCR was conducted using Prime Taq Premix (2X) (GeNet Bio, Daejeon, Korea) according to the manufacturer’s instructions. The primer sets were designed to amplify specific regions of *GmIPK1* (forward primer 5′-TCCGTTGCTTGTTGTAGCTG-3′ and reverse primer 5′-GAATGATCTGACATGAGAAG-3′) and *Bar* (forward primer 5′-ATGAGCCCAGAACGACGCCCGGCC-3′ and reverse primer 5′-GGGTCATCAGATTTCGGTGACGGG-3′). Amplification products of 718 bp and 548 bp, respectively, were expected. The inserted *SpCas9* gene was also amplified by dividing it into three parts designated as *SpCas9*-a (forward primer 5′-ATGGACAAGAAGTACAGCATCGGC-3′ and reverse primer 5′-AACTTGTAGAACTCCTCCTGGCTG-3′), *SpCas9*-b (forward primer 5′-CAGATCGGCGACCAGTACGCC-3′ and reverse primer 5′-AGAACTGGAAGTCCTTGCGGAAGT-3′), and *SpCas9*-c (forward primer 5′-CTGAGCGAGCTGGACAAGGCCGG-3′ and reverse primer 5′-TTAGGCGTAGTCGGGCACGTCGTA-3′). Amplification products of 1124 bp, 2077 bp, and 1449 bp, respectively, were expected.

### 3.6. RNA Analysis of Transgenic Plants (T_0_)

Total RNA was isolated from NT and T_0_ plants using Plant RNA Purification Reagent (Invitrogen, Carlsbad, CA, USA) according to the manufacturer’s instructions. To confirm gene expression in soybean plants, reverse transcriptase PCR (RT-PCR) was conducted using the SuPrimeScript RT-PCR Premix (2X) (GeNet Bio, Daejeon, Korea) according to the manufacturer’s instructions. The primer sets for *GmIPK1* (forward primer 5′-TCCGTTGCTTGTTGTAGCTG-3′ and reverse primer 5′-GAATGATCTGACATGAGAAG-3′) and *Bar* (forward primer 5′-ATGAGCCCAGAACGACGCCCGGCC-3′ and reverse primer 5′-GGGTCATCAGATTTCGGTGACGGG-3′) were used. The constitutive *TUB* gene (forward primer 5′-TGAGCAGTTCACGGCCATGCT-3′ and reverse primer 5′-CTCGGCAGTGGCATCCTGGT-3′) was used as the internal control to normalize the amount of RNA [38].

### 3.7. Selection of Gmipk1 Gene-Edited Soybean Plants and Generation Advance

An herbicide assay was performed to determine whether the selectable marker *Bar* gene was knocked out in *Gmipk1* gene-edited soybean plants. NT and T_1_ seeds were planted in a seedling tray, and the upper surface of two trifoliate leaves was coated across the midrib with 100 mg L^−1^ DL-phosphinothricin (PPT) mixed with Tween-20 using a brush. The response to the herbicide was observed 3–5 days after PPT application. Herbicide-sensitive and herbicide-resistant plants were separately grown in a greenhouse to obtain T_2_ seeds.

### 3.8. Determination of PA Content

Measurement of total PA in mature seeds of the T_2_ and nontransgenic lines was performed using the Megazyme PA/Total phosphorus kit (Megazyme Inc., Bray, Ireland) according to the manufacturer’s protocol. Briefly, the mature seeds were ground to a fine powder, and extraction was performed using 0.66 N hydrochloric acid with vigorous stirring for 10–12 h at ambient temperature. This method includes acid extraction of inositol phosphates and further treatment with phytase and alkaline phosphates to free phosphates. The supernatant was used for the colorimetric assay according to the manufacturer’s instructions.

## 4. Conclusions

We developed soybean gene-edited plants with a low PA content using the CRISPR/Cas9 system to mutate the *GmIPK1* gene. The mutation of *GmIPK1* led to a reduction in PA content in soybean seeds but did not show any growth defects or seed viability in soybean plants. Furthermore, identifying the superior alleles of *GmIPK1* would represent valuable resources for the genetic improvement of soybean plants with low PA levels.

## Figures and Tables

**Figure 1 ijms-23-10583-f001:**
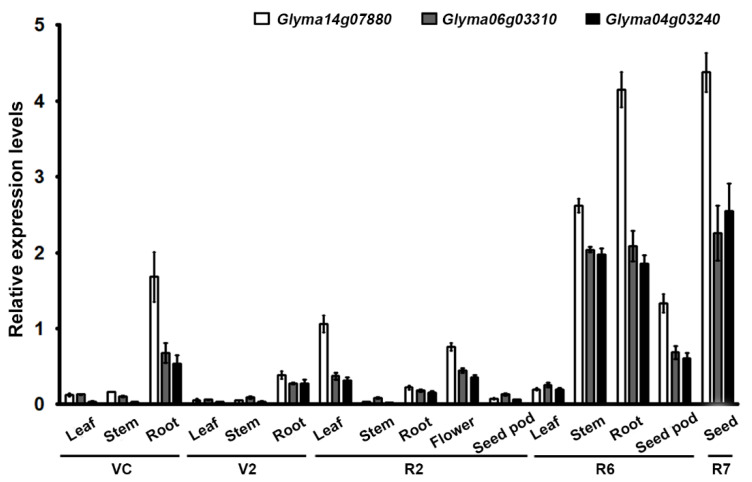
Expression analysis of *GmIPK1* (*Glyma14g07880*) (white bar) and its homologous genes *Glyma06g03310* (gray bar) and *Glyma04g03240* (black bar) in different tissues and organs of soybean (cv. Kwangankong). Relative expression levels were measured by qRT-PCR analysis following normalization with *GmPEPCo* gene control. The data are the mean of three technical replicates corresponding to each biological replicate (*n* = 3), with error bars representing standard deviations (SD). For qRT-PCR analysis, each sample was harvested at two vegetative stages (VC and V2) and three reproductive stages (R2, R6, and R7) of soybean (cv. Kwangankong) plants.

**Figure 2 ijms-23-10583-f002:**
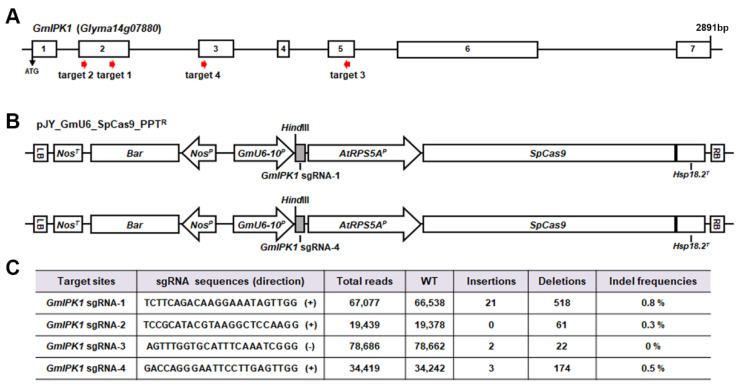
CRISPR/Cas9-targeted mutagenesis of *GmIPK1* in soybean protoplasts. (**A**) Schematic diagram of *GmIPK1* gene structure and four guide RNA target sites used in this study (red arrows). Based on a comparison with *GmIPK1* cDNA, the coding regions (4158 bp) of the *GmIPK1* gene are divided into seven exons interrupted by six introns. (**B**) The vector pJY_GmU6-10_SpCas9_ *Bar* was used for CRISPR/Cas9-mediated soybean gene editing with guide RNAs: *AtRPS5A^P^, Arabidopsis AtRPS5A* promotor; *GmU6-10^P^*, soybean *U6-10* promotor; sgRNA, single guide RNA; *Nos^P^*, nopaline synthase promoter; *Nos^T^*, nopaline synthase terminator; *SpCas9*, human codon-optimized *S. pyogenes Cas9*; NLS, nuclear location signal; *Bar*, selective marker gene. (**C**) Four sgRNA sequences of *GmIPK1* are shown. Mutation rates tested in soybean protoplasts were determined by indel frequencies (%) through deep sequencing at target regions of the *GmIPK1* gene.

**Figure 3 ijms-23-10583-f003:**
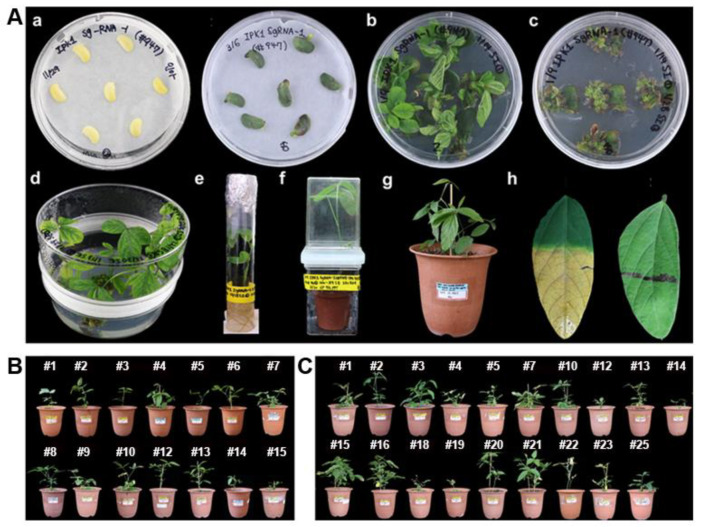
Production of *Gmipk1* gene-edited soybean transgenic plants via *Agrobacterium*-mediated transformation. (**A**) Generation of CRISPR/Cas9-mediated *Gmipk1* gene-edited transgenic soybean plants. (a) Cocultivation of half-seeds after inoculation stage (left) and at 5 days after inoculation (right). (b) Shoot induction on shoot induction medium (SIM) without PPT selection for 14 days. (c) Shoot induction on SIM with 10 mg L^−1^ PPT for another 14 days. (d) Shoot elongation on shoot elongation medium (SEM) with 5 mg L^−1^ PPT selection. (e) Root formation on rooting medium (RM). (f) Acclimation of a putative transgenic plant in a small pot. (g) Transgenic plant (T_0_) grown in a greenhouse. (h) Leaf coating with 100 mg L^−1^ PPT showing sensitivity in the leaf of the NT plant (left) and resistance in the leaf of the transgenic plant (right). (**B**) *GmIPK1* sgRNA-1 transgenic soybean plants (T_0_) grown in the greenhouse. (**C**) *GmIPK1* sgRNA-4 transgenic soybean plants (T_0_) grown in the greenhouse.

**Figure 4 ijms-23-10583-f004:**
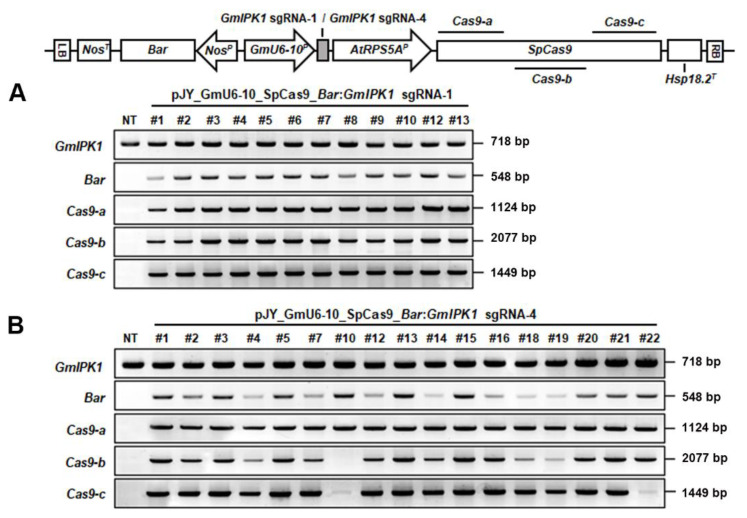
Confirmation of introduced genes from *Gmipk1* gene-edited soybean transgenic plants (T_0_) by PCR analysis. Genomic DNA was extracted from *GmIPK1* sgRNA-1 (**A**) and *GmIPK1* sgRNA-4 (**B**) T_0_ transgenic leaves to confirm the integration of *GmIPK1* and *Bar* genes. The inserted *SpCas9*, divided into three parts (*SpCas9*-a, *SpCas9*-b, and *SpCas9*-c), was analyzed by PCR: NT, nontransgenic plant; (**A**) *GmIPK1* sgRNA-1 transgenic lines (#1–#10, #12, #13); (**B**) *GmIPK1* sgRNA-4 transgenic lines (#1–#5, #7, #12–#16, #18–#22).

**Figure 5 ijms-23-10583-f005:**
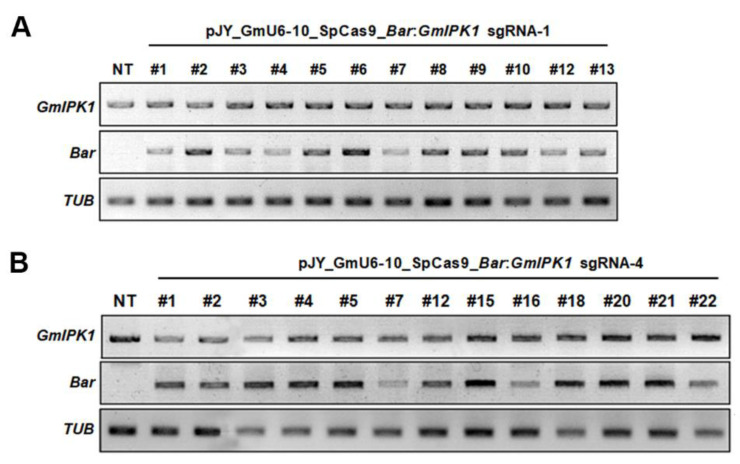
Analysis of transcript levels of introduced genes in *Gmipk1* gene-edited soybean plants (T_0_) using RT-PCR. Total RNA was extracted from *GmIPK1* sgRNA-1 and *GmIPK1* sgRNA-4 transgenic soybean plants (T_0_), and RT-PCR was used to confirm the transgene expression. *GmTUB* gene was used as a quantitative control: NT, nontransgenic plant; (**A**) *GmIPK1* sgRNA-1 transgenic lines (#1–#10, #12, #13); (**B**) *GmIPK1* sgRNA-4 transgenic lines (#1–#5, #7, #12, #15–#16, #18, #20–#22).

**Figure 6 ijms-23-10583-f006:**
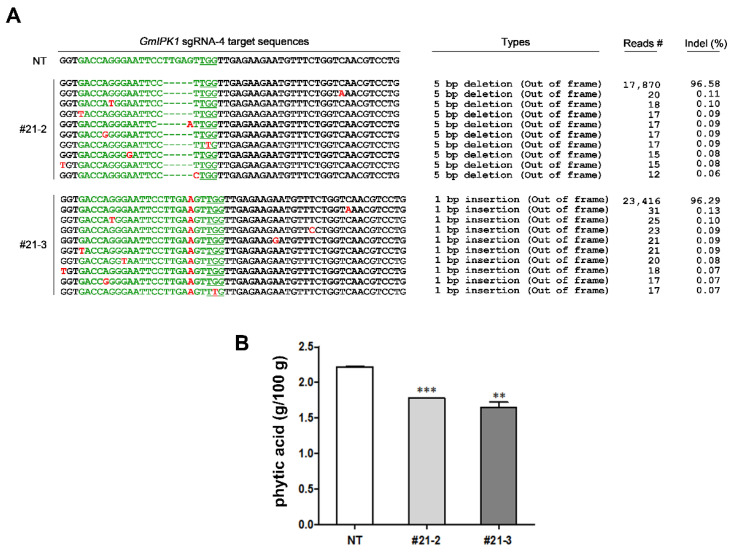
Inheritability of CRISPR/Cas9-induced mutations in *Gmipk1* gene-edited transgenic soybean plants. (**A**) Indel mutation patterns and ratios in T_1_ progenies of T_0_ plant. The mutation ratio (Indel %) was calculated by dividing the number of reads containing indels at the target site (Reads #) by the number of total sequencing reads. The protospacer adjacent motif (PAM) sequences (NGG) are bold and underlined. Insertions and deletions are represented by red font and green hyphens, respectively. (**B**) Total phytic acid contents in T_2_ seeds of soybean *Gmipk1* gene-edited lines #21-2 and #21-3. Phytic acid was measured in the mature T_2_ seeds from each GE line #21-2 and #21-3. The symbols ** and *** indicate significant differences at *p* = 0.001 and 0.0001, respectively (*n* = 3).

## Data Availability

All datasets for this study are included in the manuscript and the Appendix A.

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
