# Peer review of "Mutation of GmIPK1 Gene Using CRISPR/Cas9 Reduced Phytic Acid Content in Soybean Seeds"

_ijms, 2022, doi:10.3390/ijms231810583_

Round 1
Reviewer 1 Report
Authors describe in their article CRISPR/Cas9-mediated inactivation of GmIPK1 gene leading to lower levels of phytic acid in soybean seeds.
Experiments are well designed and correctly performed. Methods are described in sufficient details to be reproduced.
Comments:
-authors claim that “…GmIPK1 exhibited a significantly higher expression…” compared to its two homologues (lines 239, 244). However, the difference is not so convincing (max 2-times based on Fig. 1 data). Moreover, comparing PCRs using different primers is possible only when PCR efficiencies are similar – observed difference (~ 1 cycle) could be easily caused just by differences in PCR efficiencies. So, PCR efficiency analysis should be done to confirm observed results/differences and the statements/claims should be corrected accordingly.
-similarly, the claim “…expression level of GmIPK1 was significantly higher in the seed…” (line 244) is based on a normalization to Actin housekeeping gene, but such a comparison makes sense only when Actin expression is constant in all tissues/conditions tested, which is not much probable. Using (average of) more housekeeping genes is highly recommended for such an analysis. Moreover, differences in Actin and GmIPK1 gene expression levels are quite big (~ 10-1000-times, estimation based on Fig. 1, which corresponds up to 10 cycles difference on qPCR) – I would recommend to include 2-3 housekeeping gene(s) with similar expression level to GmIPK1 to minimize artifacts introduced by the normalization (you can check for example here: https://www.gene-quantification.de/chapter-3-pfaffl.pdf).
-lines 328-331 are not clear: “…must be from the endogenous or introduced genes.” is confusing since no GmIPK1 transgene was introduced to cells. PCR analyzed genomic DNA which should be present as no deletion/insertion of a whole gene was performed – just point mutations (or small ins/dels). Also statement “…missing region of the SpCas9 gene (4,170 bp)” is not clear to me what does it mean.
-copy number analysis section (Fig. S1, lines 339-346) is not much convincing. The quality of Fig. S1 is rather poor and many clones have no visible band there albeit the transgene should be present according to PCR analysis (Fig. 4). Moreover, the presence of a transgene is not much important for the study, because it has no direct connection to the presence of mutation/modification in GmIPK1 gene. So, the section regarding the presence of the transgene could be rephrased and even shorten substantially.
-decrease of PA levels in transgenic plants is not huge (just 20-25%). A possibility that GmIPK1 homologous genes could be responsible for remaining PA production is discussed and proposed combined inhibition of all isoforms may decrease PA levels more substantially. Is there any (predicted) limit how PA levels could be decreased? Is PA by itself necessary for plant development? And could PA levels be decreased even more with combined inhibition of more enzymes acting in different place(s) within PA synthesis pathway?
-using described procedure, all mutant plants will ultimately contain randomly integrated transgene(s) (expressing Cas9 and Bar) in their genomes. Such integration(s) could disrupt expression of different gene(s), even potentially important for plant development. Is there any approach available in plants to make transgene expression only transient and select mutated plants without integration of a transgene? If there is such a possibility, it should be discussed in more details.
Minor points:
-lines 70-71: “…leading to a deficiency in available P…” – probably meaning P deficiency in humans consuming soybean seeds, not deficiency in plants themselves. Should be rephrased to make it clearer.
-Clone #13 (Fig. 5B) is lacking GmIPK1 expression – was this clone analyzed in more details? What is the basis of missing expression (as RNA should be expressed regardless of the presence of mutation(s))?
-line 309: Fig 3C should be probably mentioned instead of 2C
-check font size in lines 18, 129, and 252
-correct lines 357-359: The sgRNA-1 and sgRNA-4 induced indel mutations in GmIPK1 at efficiencies ranging from 0.1-84.3% and 0.3-82.5%, respectively.
-typos: “third exon” in line 20, missing space in lines 66 and 381, “GmIPK1” and “Gmipk1” are used throughout the article – are there some reasons for different spelling (capital letters)? Please, double check the format(s).
Author Response
Thank you very much for your hard work on the submitted manuscript. According to your kind suggestions and comments, we have made complete revisions for the whole paper (shown in red color), including text content, typo errors, and so on.
Comments and suggestions for authors:
Authors describe in their article CRISPR/Cas9-mediated inactivation of GmIPK1 gene leading to lower levels of phytic acid in soybean seeds.
Experiments are well designed and correctly performed. Methods are described in sufficient details to be reproduced.
Comment 1 & 2:
- Authors claim that “…GmIPK1 exhibited a significantly higher expression…” compared to its two homologues (lines 239, 244). However, the difference is not so convincing (max 2-times based on Fig. 1 data). Moreover, comparing PCRs using different primers is possible only when PCR efficiencies are similar – observed difference (~ 1 cycle) could be easily caused just by differences in PCR efficiencies. So, PCR efficiency analysis should be done to confirm observed results/differences and the statements/claims should be corrected accordingly.
- Similarly, the claim “…expression level of GmIPK1 was significantly higher in the seed…” (line 244) is based on a normalization to Actin housekeeping gene, but such a comparison makes sense only when Actin expression is constant in all tissues/conditions tested, which is not much probable. Using (average of) more housekeeping genes is highly recommended for such an analysis. Moreover, differences in Actin and GmIPK1 gene expression levels are quite big (~ 10-1000-times, estimation based on Fig. 1, which corresponds up to 10 cycles difference on qPCR) – I would recommend to include 2-3 housekeeping gene(s) with similar expression level to GmIPK1 to minimize artifacts introduced by the normalization (you can check for example here: https://www.gene-quantification.de/chapter-3-pfaffl.pdf).
â–¶ Response
As the reviewer’s comment, we have performed qRT-PCR to check the expression of GmIPK1 homologous genes using the phosphoenolpyruvate carboxylase gene (GmPEPCo) as another housekeeping gene with a similar expression level to GmIPK1 (0.1~4-times) in soybean. Similar to the soybean Actin gene as an internal control, in our current study with GmPEPCo gene, we observed the highest expression levels of GmIPK1 homologs in the stems and roots of the R6 stage, and the seeds of the R7 stage.
Many studies have reported the use of GmPEPCo as an internal control to check the expression levels of target genes in soybean (Kumar et al. 2019; Basak et al. 2020). For example, Kumar et al. (2019) analyzed the GmMIPS1 expression together with the GmPEPCo gene as an internal control in transgenic soybean plants generated by antisense and RNAi methods. Basak et al. (2020) also used the GmPEPCo gene as an internal control to compare transcript levels of three GmIPK1 homologs in soybean.
According to the reviewer’s suggestion, thus, we used the GmPEPCo gene as an internal control to analyze the expression levels of GmIPK1 homologs in soybean. And we changed Figure 1 in the revised version of our manuscript.
References:
- Basak, N., Krishnan, V., Pandey, V., Punjabi, M., Hada, A., Marathe, A., Jolly, M., Palaka, B.K., Ampasala, D.R., and Sachdev, A. Expression profiling and in silico homology modeling of Inositol pentakisphosphate 2-kinase, a potential candidate gene for low phytate trait in soybean. 3 Biotech. 2020, 10, 1-21.
- Kumar, A., Kumar, V., Krishnan, V., Hada, A., Marathe, A., Jolly, M., and Sachdev, A. Seed targeted RNAi-mediated silencing of GmMIPS1 limits phytate accumulation and improves mineral bioavailability in soybean. Scientific Reports. 2019, 9, 1-13.
Comment 3:
Lines 328-331 are not clear: “…must be from the endogenous or introduced genes.” is confusing since no GmIPK1 transgene was introduced to cells. PCR analyzed genomic DNA which should be present as no deletion/insertion of a whole gene was performed – just point mutations (or small ins/dels). Also statement “…missing region of theSpCas9 gene (4,170 bp)” is not clear to me what does it mean.
â–¶ Response:
As you pointed out, there is no GmIPK1 as a transgene. Thus, we corrected the sentence properly and replaced Figure 4B with a newly edited one with a proper description in the revision version of the manuscript as follows: “To confirm the integration of transgenes in transgenic soybean plants, genomic DNA was isolated from 12 healthy and well-grown GmIPK1 sgRNA-1 T0 plants and 16 GmIPK1 sgRNA-4 T0 plants and analyzed via PCR (Fig. 4).”
“The GmIPK1 region, Bar gene, and SpCas9 gene were also verified in all 16 transgenic lines from GmIPK1 sgRNA-4 (Fig. 4B). Successful amplification of the GmIPK1 region (718 bp) must be from the endogenous GmIPK1 gene.”
Comment 4:
Copy number analysis section (Fig. S1, lines 339-346) is not much convincing. The quality of Fig. S1 is rather poor and many clones have no visible band there albeit the transgene should be present according to PCR analysis (Fig. 4). Moreover, the presence of a transgene is not much important for the study, because it has no direct connection to the presence of mutation/modification in GmIPK1 gene. So, the section regarding the presence of the transgene could be rephrased and even shorten substantially.
â–¶ Response:
As the reviewer’s comment, we removed Figure S1 from the Supplementary materials and revised the description of that part briefly as follows: “The copy number of transgene insertions in transgenic soybean plants (T0) was determined by Southern blot analysis of transgenic lines with sufficient leaf samples using the Bar probe. Four transgenic lines (#6, #10, #12, and #13) seem to have a single insertion, and 5 transgenic lines (#2, #3, #4, #5, and #7) have multiple copies of the transgene from the result of GmIPK1 sgRNA-1. Among GmIPK1 sgRNA-4 transformants, 7 transgenic lines (#1, #2, #5, #13, #20, #21, and #22) seem to have low copy number of transgene (data not shown).”
Comment 5:
Decrease of PA levels in transgenic plants is not huge (just 20-25%). A possibility that GmIPK1 homologous genes could be responsible for remaining PA production is discussed and proposed combined inhibition of all isoforms may decrease PA levels more substantially. Is there any(predicted) limit how PA levels could be decreased? Is PA by itself necessary for plant development? And could PA levels be decreased even more with combined inhibition of more enzymes acting in different place(s)within PA synthesis pathway?
â–¶ Response
We think there is a limit for PA levels to be decreased by the combined inhibition of PA biosynthetic genes. As you mentioned, we also think that PA by itself is necessary for plant growth and development. In addition, many studies reported that PA is crucial for cell signaling, growth, development, and plants' defense response against biotic and abiotic stresses (Raboy 2009; Kumar et al. 2021). Until now, there are two research papers showing low phytic acid soybean plants using transgenic technology. Kumar et al. (2019) generated RNAi-mediated silencing of GmMIPS1 lines with 38.75% (antisense) and 41. 34% (RNAi) reduction in PA level without compromised growth and seed development. Punjabi et al. (2018) showed that transgenic GmIPK2 silencing lines (lpa) reduced 42-45% PA level with normal morphologies and agronomic traits. GmMIPS1 enzyme plays role in the first step of PA biosynthetic pathway and GmIPK2 enzyme functions in the former step of GmIPK1 for PA biosynthesis. Based on these results, we speculate that mutations in two or three genes such as GmMIPS1, GmIPK2, and GmIPK1 could reduce PA levels by around 50% in seeds without any growth and development defects.
References:
- Raboy, V. Approaches and challenges to engineering seed phytate and total phosphorus. Plant Sci. 2009, 177, 281-296.
- Kumar, A., Singh, B., Raigond, P., Sahu, C., Mishra, U.N., Sharma, S., and Lal, M.K. Phytic acid: Blessing in disguise, a prime compound required for both plant and human nutrition. Food Res. Int. 2021, 142, 110193.
- Kumar, A., Kumar, V., Krishnan, V., Hada, A., Marathe, A., Jolly, M., and Sachdev, A. Seed targeted RNAi-mediated silencing of GmMIPS1 limits phytate accumulation and improves mineral bioavailability in soybean. Scientific Reports. 2019, 9, 1-13.
- Punjabi, M., Bharadvaja, N., Jolly, M., Dahuja, A., and Sachdev, A. Development and evaluation of low phytic acid soybean by siRNA triggered seed specific silencing of inositol polyphosphate 6-/3-/5-kinase gene. Front. Plant Sci. 2018, 9, 804.
Comment 6:
Using described procedure, all mutant plants will ultimately contain randomly integrated transgene(s) (expressing Cas9 and Bar) in their genomes. Such integration(s) could disrupt expression of different gene(s), even potentially important for plant development. Is there any approach available in plants to make transgene expression only transient and select mutated plants without integration of a transgene? If there is such a possibility, it should be discussed in more details.
â–¶ Response
There are some experimental methods to make transgene expression transiently in plants without integration of any transgene. In plants, ribonucleoprotein (RNP) or plasmids harboring the Cas and sgRNA sequences can be introduced directly into protoplasts using transient transfection, allowing recombinant DNA-free gene-editing plants to be regenerated. However, plant regeneration from protoplasts remains unestablished in many plant species, being especially difficult and limited to only some crops, such as tomato, tobacco, and lettuce. Until now, in soybean, there is no report on the application of DNA-free gene-editing method using protoplast regeneration. Even though, it is still difficult to develop soybean transgenic plants with a binary vector by agrobacterium-mediated transformation. Thus, it is highly required to set up soybean regeneration methods from protoplasts to whole plants for RNP-mediated gene editing.
Reviewer 2 Report
The current version is not accepted for publication, major revision needed, see editor comments below
1. Please update your introduction and discussion sections with more reviews 2018, 2019-2020 and 2021.
2. Separate results and discussion into two parts
3. Article objectives is not clear.
4. Many sentences do not have the correct punctuation and it is difficult to read the text.
5. English should be improved; grammar need for enhancement in many sentences and paragraphs.
6. All figures need for resolution enhancement.
7. Figures is not in printable quality. Also, some portions of the texts are losing their readability while sizing the image as per text area. Kindly provide a better-quality figure.
8. Please check the References in-text and end-list for uniformity in style.
9. The conclusion is needed in the end of the discussion part
Reformat your article based on the MDPI format
Author Response
Comment 1:
Please update your introduction and discussion sections with more reviews 2018, 2019-2020 and 2021.
â–¶ Response
As the reviewer’s comment, we updated the references from the Introduction and Results & Discussion sections.
Updated references:
- Jin, H., Yu, X., Yang, Q., Yuan, F. and Fu, X. Transcriptome analysis identifies differentially expressed genes involved in the metabolic regulatory network of progenies from the cross of low phytic acid GmMIPS1 and GmIPK1 soybean mutants. Sci. Rep. 2021, 11, 8740
- Kumar, A., Singh, B., Raigond, P., Sahu, C., Mishra, U.N., Sharma, S., and Lal, M.K. Phytic acid: Blessing in disguise, a prime compound required for both plant and human nutrition. Food Res. Int. 2021, 142, 110193.
- Raboy, V. Low phytic acid crops: Observations based on four decades of research. Plants, 2020, 9, 140.
Comment 2:
Separate results and discussion into two parts.
â–¶ Response
In the IJMS’s policy, they descript “Discussion section may be combined with Results section”. Two separate sections of Results and Discussion are not mandatory. Thus, we prefer to keep the Results and Discussion as one combined section.
Comment 3:
Article objectives is not clear.
â–¶ Response
As you commented, we added the research objectives at the end of the Introduction and Results & Discussion sections.
Comment 4:
Many sentences do not have the correct punctuation and it is difficult to read the text.
â–¶ Response
During the conversion from our MS word form to the IJMS one, we think that the problems occurred. Anyway, we corrected sentence structure and punctuation marks in the revised version of our manuscript.
Comment 5:
English should be improved; grammar need for enhancement in many sentences and paragraphs.
â–¶ Response
We have proofread our manuscript by a professional English proofreading service (Editage) before submission. Also, we read carefully and corrected the manuscript according to the editor and reviewers’ comments.
Comment 6 & 7:
- All figures need for resolution enhancement.
- Figures is not in printable quality. Also, some portions of the texts are losing their readability while sizing the image as per text area. Kindly provide a better-quality figure.
â–¶ Response
During the conversion from our MS word form to the IJMS one, we think that the problems occurred. As you commented, we improved all figures with high resolution.
Comment 8:
Please check the References in-text and end-list for uniformity in style.
â–¶ Response:
Thanks. We corrected the references in-text and end-list style.
Comment 9:
The conclusion is needed in the end of the discussion part Reformat your article based on the MDPI format.
â–¶ Response
As the reviewer’s comment, we added the Conclusion section at the end of Results and Discussion section in the revised version of our manuscript.
Round 2
Reviewer 2 Report
Accept in present form
